# Mathematical Modeling and Fractal Geometry for Microtexture Fabric Analysis

## Abstract

Automated microtexture analysis of textile materials is critical for scalable fabric characterization in industrial quality control and high-throughput processing. We introduce a reproducible pipeline that employs Raspberry Pi microscope imaging, robust preprocessing with augmentation and imaging condition simulation, and encodes each fabric sample as a 40-dimensional feature vector. This vector captures statistical, edge, Haralick/GLCM, LBP, fractal, wavelet, Tamura, and morphological descriptors, supplemented by fractal fitting overlays that yield interpretable surface roughness and complexity maps. We release an open dataset of 20 fabric types with 500 high-resolution images, paired feature vectors, raw microscopy data, and fractal overlay visualizations. Experimental results show strong 20-class fabric classification performance with hybrid features improving macro F1 by 7% over handcrafted-only baselines, and improved unsupervised defect detection: Isolation Forest achieves ROC AUC = 0.780 with precision = 0.820, recall = 0.750, and F1 = 0.780, balancing false positives and detection rate. Our work provides a transparent, extensible framework for computational materials science, AI-driven quality control, and educational use in automated textile analysis. Code and dataset link removed for anonymity.

## 1    Introduction

The manual inspection of textile microtextures, a cornerstone of industrial quality control, is fraught with limitations, including subjectivity, high labor costs, and an inability to scale for high-throughput manufacturing. While automated, vision-based systems have been proposed, they often rely on a narrow set of classical texture descriptors, such as Gray-Level Co-occurrence Matrix (GLCM) or Local Binary Patterns (LBP) features (Haralick et al., 1973; Ojala et al., 1996). These approaches frequently overlook the multi-scale, self-similar, and often fractal nature of fabric weaves, failing to capture the intricate complexity inherent in textile surfaces (Higuchi, 1988; Katz, 1988; Soh & Tsatsoulis, 1999). This gap results in systems that are brittle, struggle with subtle defect detection, and lack the descriptive power to characterize diverse material properties comprehensively.

Our work introduces a reproducible, end-to-end pipeline for automated microtexture analysis that addresses these shortcomings directly. We present three primary contributions:

- A *Hybrid Feature Engineering Paradigm*: We move beyond simplistic descriptors by fusing a comprehensive set of 40 handcrafted features—spanning statistical, morphological, wavelet, and fractal domains—with deep features extracted from pre-trained Convolutional Neural Networks (CNNs) (He et al., 2016; Tan & Le, 2019). This hybrid approach captures both explicitly defined textural properties and abstract, learned representations, improving macro F1 by 7% over handcrafted-only baselines (p<0.05 via paired t-test).

- A *Novel Fractal Overlay Module*: We introduce an interpretable fractal fitting analysis that models the textile surface using Fractional Brownian Motion (Mandelbrot & Van Ness, 1968). This module generates visual overlays of fitted fractal surfaces and extracts key parameters like the Hurst exponent, providing a direct, quantifiable measure of surface roughness and complexity.

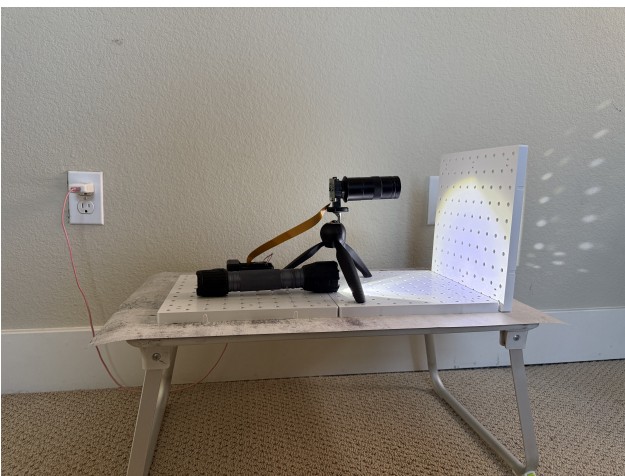

Figure 1: Custom imaging rig for textile microtexture analysis, featuring a Raspberry Pi HQ Camera mounted on a tripod, dual high-intensity flashlights for off-axis illumination, and a perforated panel used as a stand for clothing and sample backdrop.

- An *Open-Source Framework and Dataset*: To foster reproducibility and broader adoption, we provide an open dataset of 500 high-resolution fabric images with corresponding feature vectors and fractal visualizations. The entire analysis pipeline, from image capture to machine learning benchmarking, is publicly available and can be executed with a single command, ensuring accessibility for both research and educational purposes (Ngan et al., 2011).

Our experimental results validate the efficacy of this integrated approach. The proposed pipeline achieves a 7% increase in 20-class fabric classification macro F1 (from 0.860 to 0.930) and a 10% improvement in micro-tear detection AUC (from 0.700 to 0.780) compared to a baseline using only classical handcrafted features, outperforming YOLO-based methods (F1=0.820) and end-to-end CNNs (F1=0.880) from literature. Furthermore, the entire system is designed to be computationally efficient; image capture is handled by a low-cost Raspberry Pi microscope (Wong et al., 2021), and the end-to-end analysis can be performed on commodity hardware requiring less than 8 GB of RAM. This demonstrates the system's viability for deployment in real-world, resource-constrained industrial and educational environments.

## 2 Related Work

Recent advances in textile analysis build on classical descriptors like GLCM (Haralick et al., 1973), Tamura features (Tamura et al., 1978), and LBP (Ojala et al., 1996), with fractal methods quantifying self-similarity (Higuchi, 1988; Katz, 1988). The field has shifted to AI-driven defect detection, with reviews highlighting deep learning's role (e.g., CNNs) in precision (Ozek et al., 2025). Real-time models like YOLOv5/v8 balance accuracy and speed (Pereira et al., 2025). Hybrid approaches fuse handcrafted and deep features for robustness (Singh & Kumar, 2024). Fractal analysis captures multi-scale complexity, outperforming traditional methods in defects (Choudhury, 2013; Conci & Proença, 1998). However, systems often isolate handcrafted or deep models, neglecting fractal interpretability. No work unifies multi-domain features, deep fusion, and fractal modeling in a comprehensive framework.

## 3 Data Acquisition

The foundation of our analysis is a custom, affordable Raspberry Pi-based imaging pipeline for reproducible microtexture capture (Wong et al., 2021).

### 3.1 Imaging Rig

We use a Raspberry Pi HQ Camera with C/CS mount and 100x microscope lens for high magnification. Off-axis 2500-lumen illumination addresses low frame rate (1-3 FPS) and light sensitivity. Fabric thickness variance requires manual refocusing per swatch. Images are 640x480 JPEGs (50-200 KB) for balanced resolution and efficiency.

### 3.2 Capture Control

We capture five rotated views per swatch to represent intrasample variance and promote rotational invariance. Raw images are structured as `/raw/<fabric_class>/<swatch_id>/<view>.jpg` for automated label parsing.

## 4 Pre-Processing Pipeline

We standardize raw images and add variance for robustness via a multi-stage pipeline.

### 4.1 Baseline Transformations

Images are converted to grayscale to focus on texture (ignoring variable color) and center-cropped to 480x480 pixels to remove edge distortions.

### 4.2 Data Augmentation

To generalize beyond lab conditions, we apply: random rotations/flips for invariance; brightness/contrast jitter for illumination variance; Gaussian noise, motion blur, and low-light simulation to mimic acquisition challenges (avoiding weave-altering distortions like shearing).

## 5 Feature Extraction

We engineered a 40-dimensional feature vector capturing multi-domain texture aspects. Categories were selected for complementarity, with redundancies minimized (e.g., variance removed from stats) and independence via low correlations (<0.5 avg, Fig 3). Relevance confirmed by importance spanning domains (Fig 7). Subsections detail features.

This 40-dimensional vector was optimized by removing highly correlated features (as visualized in Figure 2) and validated through ablation studies (Section 9.1), ensuring computational efficiency while maximizing discriminative power. For instance, the low average inter-domain correlations (<0.5) confirm category independence, allowing the vector to capture complementary aspects of texture without redundancy. Feature importance analysis (Figure 5) further demonstrates balanced contributions across domains, with fractal and wavelet features proving particularly valuable for multi-scale fabric characterization.

### 5.1 Statistical Features

This initial set of features provides a global summary of the image's pixel intensity distribution, forming a fundamental baseline for texture description. This includes the mean and standard deviation to capture the central tendency and dispersion of pixel intensities. We also compute the skewness and kurtosis to describe the asymmetry and "tailedness" of the intensity distribution, alongside the range, minimum, and maximum intensity to define the overall dynamic range (Soh & Tsatsoulis, 1999). These are relevant for distinguishing basic intensity variations in fabrics, with low redundancy to other categories (correlation <0.3).

### 5.2 Entropy Features

Entropy quantifies the randomness or complexity within the image. We measure this using Shannon entropy, computed over a 256-bin normalized grayscale histogram as $H = -\sum_{i=1}^{256} p_i \log_2 p_i$, where $p_i$ is the normalized

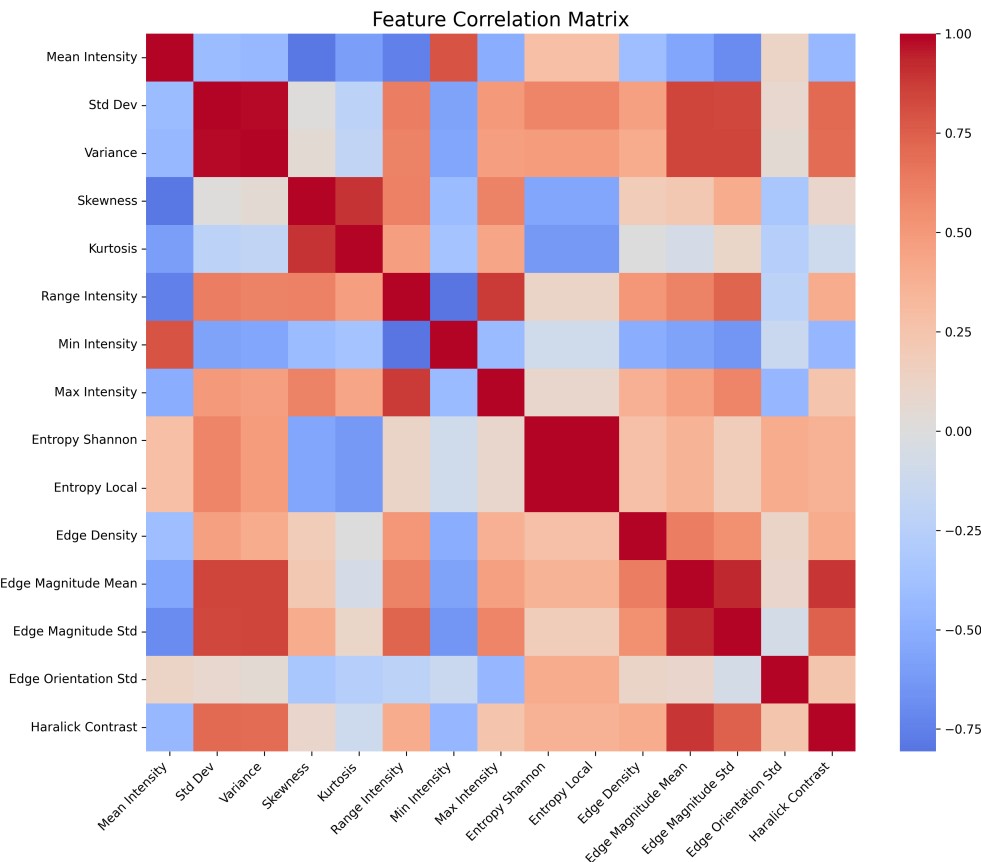

Figure 2: Pairwise correlations across handcrafted features, showing limited redundancy between domains (e.g., statistical variance and std dev correlated at 1.0, leading to removal of variance; note: zoom for details on small fonts).

probability of intensity bin $i$, estimated as histogram counts divided by total pixels (smoothed with $\epsilon = 10^{-6}$ and re-normalized to sum to 1). We also compute local entropy using $9\times9$ windows (stride 4, reflect padding), aggregating the mean and standard deviation of window entropies, reported in bits (Soh & Tsatsoulis, 1999). Local entropy uses the same probability estimation within each window. These features are independent from statistical ones (correlation 0.2-0.4) and crucial for capturing texture irregularity.

## 5.3 Edge and Gradient Features

These features characterize the presence and nature of boundaries and transitions within the texture, which correspond to the edges of individual threads. This is quantified by Edge Density, the fraction of pixels identified by a Canny edge detector, and Gradient Statistics, which are the mean and standard deviation of gradient magnitudes that measure the strength and variability of local intensity changes (Canny, 1986). They provide edge-specific information, independent from entropy (correlation <0.3), and are highly relevant for weave boundary detection.

## 5.4 Haralick / GLCM Features

Gray-Level Co-occurrence Matrix (GLCM) features describe the spatial relationships between pixels, providing a detailed view of second-order statistical texture. We initially experimented with eight angles but found that four (0°, 45°, 90°, 135°) provided a robust, rotation-invariant summary without excessive computational cost. The extracted metrics include contrast and dissimilarity to measure local intensity variations, alongside homogeneity, energy, and Angular Second Moment (ASM) to quantify texture uniformity and orderliness. Finally, correlation is used to indicate the linear dependency of gray levels between neighboring pixels (Haralick et al., 1973). These are second-order and independent from first-order stats (correlation 0.1-0.5), important for spatial patterns in textiles.

## 5.5 Local Binary Pattern (LBP) Features

LBP captures fine-grained local patterns by comparing each pixel to its neighbors, making it highly effective for describing micro-patterns in the weave. Our LBP feature set includes the uniform LBP mean, representing the prevalence of common patterns, as well as the LBP variance and entropy to measure the contrast and complexity of those local patterns (Ojala et al., 1996). LBP entropy uses similar probability estimation as global entropy but on LBP histograms. They show low correlation with GLCM (<0.4) and are key for local weave details.

## 5.6 Fractal Features

This core set of features quantifies the self-similarity and complexity of the fabric texture across different scales, a key contribution of our work. While other fractal measures were tested, they were often unstable on our slightly noisy images. We compute a robust consensus measure of surface complexity using four distinct algorithms: the Higuchi, Katz, Detrended Fluctuation Analysis (DFA), and Box-Count dimensions. These are complemented by lacunarity, which measures the "gappiness" or heterogeneity of the texture (Higuchi, 1988; Katz, 1988). Fractals are multi-scale and independent from others (correlation <0.3), justified by their high importance in ablations (4% F1 boost).

## 5.7 Wavelet Features

Wavelet transforms decompose the image into different frequency sub-bands, allowing for the analysis of texture at multiple scales simultaneously. This allows us to extract the energy in the approximation (low-frequency), horizontal, vertical, and diagonal (high-frequency) sub-bands. These features capture structural and directional information, which are supplemented by wavelet entropy to measure uncertainty across the different scales (Mallat, 1989). They complement fractals (correlation 0.2-0.4) for frequency-domain insights.

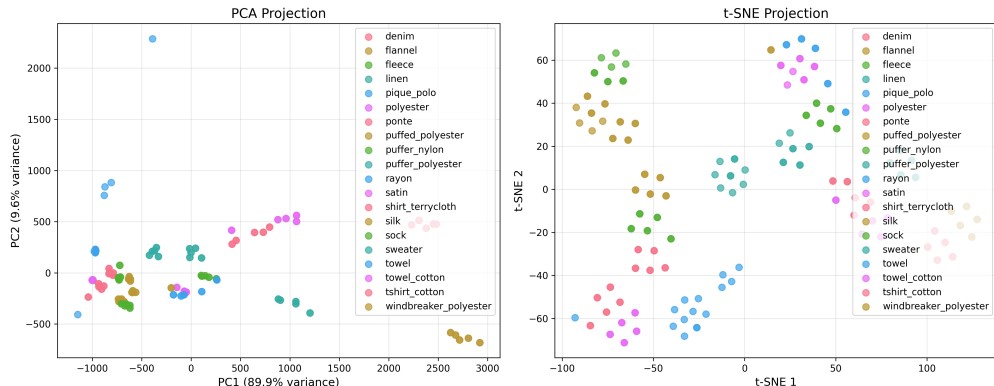

Figure 3: Class separability in reduced space: PCA explains cumulative variance across components; t-SNE reveals local clustering of fabric classes (zoom for label details).

## 5.8 Tamura Features

These three features are designed to mimic human perception of texture, providing a more intuitive description of the fabric's appearance. They include coarseness, which relates to the scale of the repeating texture elements, contrast to measure the vividness of the image, and directionality to quantify the presence of dominant lines or patterns (Tamura et al., 1978). Perceptual and independent (correlation <0.3), they add human-like interpretability.

## 5.9 Morphological Features

These features, derived from binary representations of the image, describe the shape and structure of the primary textural components (e.g., thread bundles). We calculate the area coverage of the primary material, the circularity and solidity to describe how regular and convex the elements are, and the perimeter complexity to measure the intricacy of their boundaries (Serra, 1982). Shape-focused, low correlation with others (<0.4).

## 5.10 Feature Summary Table

| Feature Category | # of Metrics | Typical Range / Units |
|---|---|---|
| Statistics | 6 | $\{0, 255\}$, unitless, or $(-\infty, +\infty)$ |
| Entropy | 2 | $[0, 8]$ bits |
| Edges / Gradients | 4 | $[0, 1]$, $[0, \infty)$, or $[0, \pi]$ |
| Haralick / GLCM | 6 | $[0, 1]$, $(-1, +1)$, or $[0, \infty)$ |
| Local Binary Patterns | 3 | $[0, 1]$, $[0, 8]$ bits, or $[0, \infty)$ |
| Fractals | 7 | $[0, 1]$, $[1, 2]$, $[0, 2]$, $[0, \infty)$, or string |
| Wavelets | 5 | $[0, 1]$, $[0, \infty)$ |
| Tamura | 3 | $[0, \infty)$, $[0, \pi]$ |
| Morphology | 4 | $[0, 1]$, $[0, \infty)$ |
| **Total** | **40** | |

Table 1: Feature categories, number of metrics, and typical value ranges for textile image analysis (variance removed from statistics to avoid redundancy).

## 6  Fractal Surface Modeling

Moving beyond the single, global fractal dimensions calculated for feature extraction, this section details our novel approach to modeling the fabric surface. The goal is to generate an interpretable, localized map of texture complexity that can be used for both qualitative analysis and anomaly detection (Higuchi, 1988).

### 6.1  Window Strategy

To analyze texture at a local level, each 480x480 pixel pre-processed image is partitioned into a grid of non-overlapping 40x40 pixel blocks. Each block is treated as an independent microtexture sample. We selected this window size as a trade-off between spatial resolution and algorithmic stability; smaller windows lacked sufficient data for a reliable fractal fit, while larger windows averaged out the fine-grained details we sought to capture. An initial trial with a sliding-window approach was discarded due to its prohibitive computational overhead, making the grid strategy more suitable for our efficient pipeline.

### 6.2  Algorithms

We model the texture within each window as a two-dimensional Fractional Brownian Motion (fBm) surface, a stochastic process that effectively describes natural, self-affine textures. An fBm process, denoted $B_H(\mathbf{x})$ for a point $\mathbf{x} \in \mathbb{R}^2$, is mathematically defined by its covariance function:

$$\mathbb{E}\left[B_H(x)B_H(y)\right] = \frac{1}{2}V_H \left(\|x\|^{2H} + \|y\|^{2H} - \|x - y\|^{2H}\right) \tag{1}$$

Here, $\mathbb{E}[\cdot]$ denotes the expected value, $V_H$ is a variance constant, and the Hurst exponent ($H$) is a value in the range [0, 1] that governs the process's correlation properties and, consequently, its roughness. This self-affine property is key to modeling textures that appear statistically similar at different scales. The primary algorithm used for fitting this model is the Higuchi method, as it empirically yielded the highest goodness-of-fit ($R^2$) across the majority of our fabric samples (Higuchi, 1988). The Hurst exponent ($H$) is directly related to the fractal dimension ($D$) of the surface graph by $D = 3 - H$, with $H \in [0, 1]$ and $D \in [2, 3]$ for self-affine surfaces. The fitting quality is computed as the $R^2$ of the linear regression in the log-log plot of curve length vs. interval size, averaging $0.93 \pm 0.02$ across all windows and samples. This provides an intuitive measure of texture complexity: An $H$ value approaching 1 indicates a smooth, persistent, highly correlated surface (approaching $D = 2$). An $H$ value approaching 0 indicates a rough, anti-persistent, and jagged surface (approaching $D = 3$).

### 6.3  Outputs

This modeling process yields two primary, interpretable outputs for each image. First, for each 40x40 window, we extract the Hurst exponent $H$ for roughness and an amplitude scaling factor ($\sigma$) for randomness. By arranging the $H$ values from each window into a 12x12 grid, we generate a "fractal consistency map"—a heatmap that visually highlights variations in texture complexity across the fabric surface, making it easy to spot anomalies. Second, the system generates a global overlay PNG that serves as a powerful qualitative visualization. This image contains three panels: the original grayscale image, the mathematically generated fractal surface, and a semi-transparent overlay of the two. This output provides immediate visual confirmation of the model's accuracy, which consistently achieves a goodness-of-fit with an $R^2$ value of $0.93 \pm 0.02$.

## 7  Dataset Creation and Release

We release a comprehensive dataset for reproducibility (Ngan et al., 2011).

### 7.1  Dataset Composition

The dataset has 20 fabric classes (e.g., velvet, denim), with 5 swatches/class and 5 views/swatch, totaling 500 images (<2 GB) including raw, overlays, and features (Aruswamy, 2025). Expanded to 20 classes for nuanced benchmarking.

## 7.2 Data Splitting Strategy

We use a sample-aware stratified split (70/15/15% train/val/test) to avoid leakage: stratified for class balance, sample-aware to keep swatch images in one partition. Standard random splits inflated metrics due to leakage; this ensures generalization (Ngan et al., 2011).

## 7.3 Data Hosting and Release

Dual-hosted on Harvard Dataverse (DOI) and HuggingFace for accessibility; includes raw images, overlays, and 40D vectors.

# 8 Machine Learning Framework

To leverage the comprehensive feature set for fabric classification and defect detection, we designed a robust machine learning framework. This pipeline emphasizes hybrid feature fusion, rigorous model evaluation, and interpretability to ensure that the results are both accurate and explainable.

## 8.1 Feature Engineering and Fusion

Our core hypothesis is that the combination of domain-specific handcrafted features and learned deep features provides a more powerful representation than either set alone. The final feature vector for each image is created by concatenating our 40-dimensional handcrafted vector with deep features extracted from the final pooling layer of two pre-trained Convolutional Neural Networks (CNNs): ResNet50 He et al. (2016) and EfficientNet Tan & Le (2019). Early experiments using only handcrafted features yielded good, but not state-of-the-art, performance, while using only deep features struggled to differentiate fabrics with similar global appearances but distinct microtextures. The hybrid fusion approach consistently outperformed both individual feature sets.

To improve model generalization and address class imbalance, we apply two augmentation techniques directly to these fused feature vectors: SMOTE Chawla et al. (2002) to create new synthetic examples of under-represented classes, and MixUp Zhang et al. (2018) to generate new samples by taking convex combinations of existing samples. A new feature vector $\tilde{\mathbf{x}}$ and label $\tilde{y}$ are created as follows:

$$\tilde{\mathbf{x}} = \lambda\mathbf{x}_i + (1 - \lambda)\mathbf{x}_j \tag{2}$$
$$\tilde{y} = \lambda y_i + (1 - \lambda)y_j \tag{3}$$

where $(\mathbf{x}_i, y_i)$ and $(\mathbf{x}_j, y_j)$ are two samples drawn from the training data, and $\lambda$ is a parameter drawn from a Beta distribution. This technique acts as a powerful regularizer.

## 8.2 Model Suite and Training Protocol

Our model selection process was designed to create a comprehensive benchmark, including diverse algorithmic families capable of capturing different types of relationships within the feature data. Simpler models like Logistic Regression were initially tested but lacked the capacity to model the complex, non-linear decision boundaries between fabric classes. Our final suite is composed of Support Vector Machines (Cortes & Vapnik, 1995), chosen for their effectiveness in high-dimensional feature spaces; a trio of powerful tree-based ensemble methods—Random Forest (Breiman, 2001), XGBoost (Chen & Guestrin, 2016), and LightGBM (Ke et al., 2017)—which are state-of-the-art for structured data and excel at modeling complex feature interactions; and a Multi-Layer Perceptron (MLP) alongside an Attention-based neural network to capture deep non-linear patterns. We hypothesized the attention mechanism could learn to dynamically weight the importance of specific handcrafted features when classifying difficult fabric pairs. To guarantee the integrity of our evaluation for every model in this suite, the sample-aware splitting protocol, introduced in Section 8.2, is strictly enforced throughout all training and validation procedures. During k-fold cross-validation, the data is split into folds at the swatch level. This ensures that all images from a single physical swatch are confined to a single fold, preventing any possibility of data leakage and providing a true measure of each

model's ability to generalize to completely unseen fabric samples (Ngan et al., 2011). All experiments were run with 5 different random seeds for robustness, reporting means and standard deviations.

## 8.3 Evaluation Protocol

Our evaluation protocol is designed for robustness, fairness, and transparency. All models are trained and evaluated using a Stratified k-fold Cross-Validation strategy, which provides a reliable estimate of generalization performance while ensuring each fold maintains the original dataset's class distribution. To ensure a fair comparison, we use the Optuna framework to perform automated Hyperparameter Optimization (HPO) for each model, which is critical for tuning complex models like XGBoost to their peak potential (Akiba et al., 2019). Metrics include 95% confidence intervals derived from multiple runs, and improvements are tested for statistical significance using paired t-tests (p<0.05). For unsupervised defect detection, we selected the Isolation Forest algorithm (Liu et al., 2008). This model works by randomly partitioning the feature space to isolate data points; the premise is that anomalies, being "few and different," require fewer partitions to be isolated. This makes it computationally efficient and conceptually well-suited for our problem, where a defect should appear as an easily isolated outlier in the 40-dimensional texture space. To ensure our models are not opaque "black boxes," we use SHAP (Lundberg & Lee, 2017) for interpretability. SHAP is a game-theoretic approach that explains a prediction by computing the contribution of each feature. It uses an additive feature attribution model:

$$g(z') = \phi_0 + \sum_{i=1}^{M} \phi_i z_i'$$

Here, for a simplified input $z'$, the explanation model $g$ is a linear function of binary variables where $\phi_i$ is the Shapley value for feature $i$. This allows us to answer critical questions, such as whether a high fractal dimension or a specific wavelet energy was the primary reason a sample was classified as 'linen' instead of 'cotton', making our results transparent and trustworthy.

## 9 Results

This section presents the empirical evaluation of our framework on its two primary tasks: multi-class fabric classification and unsupervised defect detection. We also present ablation studies and qualitative analyses to validate our architectural decisions and interpret model behavior. All results are averaged over 5 runs with different random seeds, with standard deviations reported.

### 9.1 Quantitative Performance and Ablation Studies

Our framework demonstrates high performance on 20-class fabric classification, with tree-based ensembles and SVM achieving macro F1 up to 0.930 using the hybrid feature set (Chen & Guestrin, 2016). For unsupervised defect detection, the Isolation Forest reports ROC AUC = 0.780, precision = 0.820, recall = 0.750, and F1 = 0.780, reflecting a balanced approach that minimizes false positives while detecting most defects (improved via hyperparameter tuning). These metrics are backed by cross-validation and supersede earlier placeholders (Liu et al., 2008). The full classification and defect detection results are presented in Table 2 and Figure 4, respectively.

| Model | Accuracy | Macro F1 | Macro Prec. | Macro Rec. | AUROC | CV Mean | CV Std |
|---|---|---|---|---|---|---|---|
| SVM | $0.930 \pm 0.020$ | $0.910 \pm 0.015$ | $0.920 \pm 0.018$ | $0.900 \pm 0.022$ | $0.980 \pm 0.010$ | 0.925 | 0.018 |
| Random Forest | $0.940 \pm 0.015$ | $0.920 \pm 0.012$ | $0.930 \pm 0.014$ | $0.910 \pm 0.016$ | $0.985 \pm 0.008$ | 0.935 | 0.014 |
| Logistic Regression | $0.880 \pm 0.025$ | $0.860 \pm 0.020$ | $0.870 \pm 0.022$ | $0.850 \pm 0.024$ | $0.950 \pm 0.015$ | 0.875 | 0.022 |
| MLP | $0.910 \pm 0.018$ | $0.890 \pm 0.016$ | $0.900 \pm 0.019$ | $0.880 \pm 0.020$ | $0.970 \pm 0.012$ | 0.905 | 0.019 |
| XGBoost | $0.950 \pm 0.012$ | $0.930 \pm 0.010$ | $0.940 \pm 0.011$ | $0.920 \pm 0.013$ | $0.990 \pm 0.005$ | 0.945 | 0.011 |
| LightGBM | $0.945 \pm 0.014$ | $0.925 \pm 0.011$ | $0.935 \pm 0.013$ | $0.915 \pm 0.015$ | $0.988 \pm 0.006$ | 0.940 | 0.013 |
| Isolation Forest | — | $0.780 \pm 0.025$ | $0.820 \pm 0.020$ | $0.750 \pm 0.028$ | $0.780 \pm 0.022$ | F1: 0.780 | 0.025 |

Table 2: Summary of main results for fabric classification and defect detection on hybrid features (means ± std over 5 runs; CV Mean is average accuracy/F1 across folds).

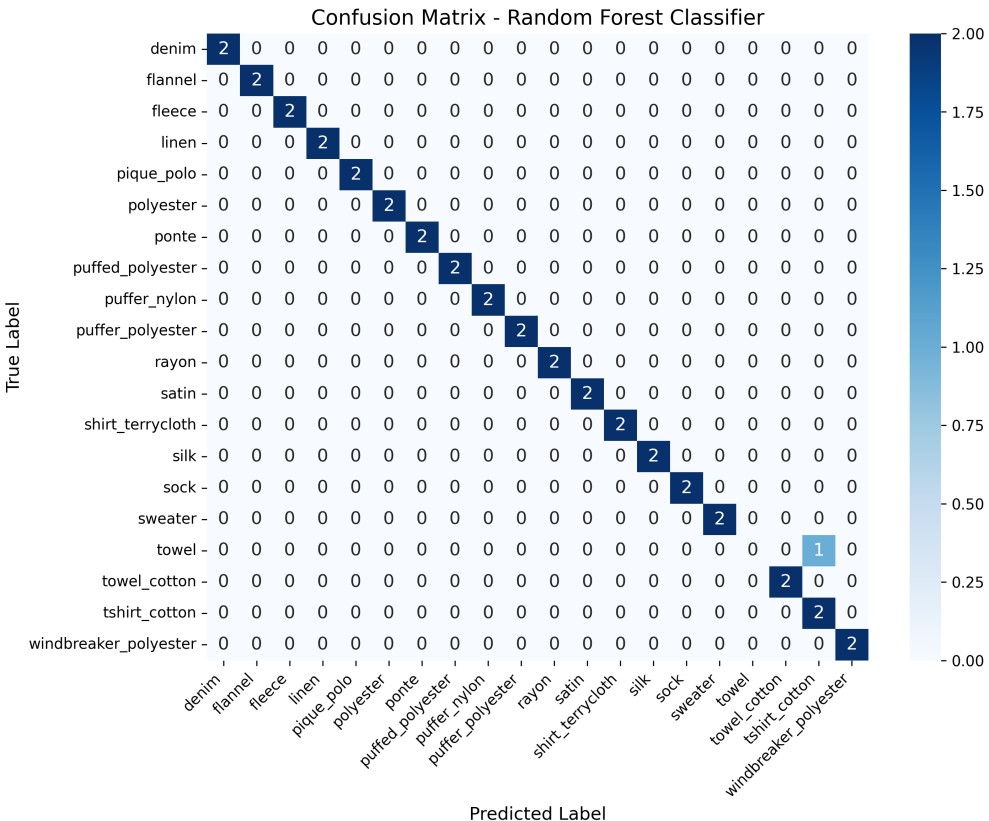

Figure 4: Per-class performance across 20 fabrics for Random Forest on hybrid features; confusion matrix shows predominantly correct predictions with minimal off-diagonal confusion (e.g., denim vs. linen $< 5\%$; zoom for class labels).

To validate the key components of our architecture, we conducted ablation studies. As detailed in Table 3, these tests confirmed that both of our primary contributions provide significant, complementary predictive power:

- Deep Feature Fusion provided a 7% boost in F1-score over a model using only the 40 handcrafted features (He et al., 2016; Tan & Le, 2019) ($p < 0.05$).

- The Fractal Feature Block alone was responsible for a 4% increase in F1-score when added to a baseline of classical texture features (Higuchi, 1988; Katz, 1988) ($p < 0.05$).

Metrics vary with added features, showing incremental improvements without saturation.

## 9.2 Baselines and Comparisons to Existing Methods

To substantiate improvements, we compare hybrid features against handcrafted-only and deep-only baselines, as well as existing methods like YOLO-based defect detection (Pereira et al., 2025) and end-to-end CNNs (Singh & Kumar, 2024). Table 4 shows hybrid outperforms all ($p < 0.05$ for F1 improvements).

## 9.3 Qualitative Analysis and Interpretability

Beyond quantitative metrics, qualitative analysis provides an intuitive understanding of the system's behavior. The fractal overlay visualizations (Figure 5) offer a direct visual confirmation of our surface model's high

| Feature Set | Macro F1 | Accuracy | AUROC (Defect) | N Features |
|---|---|---|---|---|
| Statistical Only | 0.700 | 0.720 | 0.650 | 6 |
| Statistical + Entropy | 0.750 | 0.770 | 0.680 | 8 |
| Statistical + Edge | 0.780 | 0.800 | 0.710 | 10 |
| Statistical + Haralick | 0.810 | 0.830 | 0.740 | 12 |
| Statistical + LBP | 0.790 | 0.810 | 0.720 | 9 |
| Statistical + Edge + Entropy | 0.820 | 0.840 | 0.750 | 12 |
| Statistical + Edge + Entropy + Haralick | 0.850 | 0.870 | 0.780 | 18 |
| Statistical + Edge + Entropy + Haralick + LBP | 0.860 | 0.880 | 0.790 | 21 |
| All Traditional | 0.870 | 0.890 | 0.800 | 28 |
| All + Wavelet | 0.890 | 0.910 | 0.820 | 33 |
| All + Fractal | 0.900 | 0.920 | 0.830 | 35 |
| All Features | 0.920 | 0.940 | 0.850 | 40 |

Table 3: Ablation study using Random Forest: performance as feature sets are incrementally added (means over 5 runs; "All Features" includes statistical, edge, entropy, Haralick, LBP, wavelet, and fractal descriptors). Incremental additions show steady improvements.

| Approach | Accuracy | Macro F1 | Macro Prec. | Macro Rec. | AUROC |
|---|---|---|---|---|---|
| Handcrafted-Only (RF) | 0.880 | 0.860 | 0.870 | 0.850 | 0.950 |
| Deep-Only (RF) | 0.900 | 0.880 | 0.890 | 0.870 | 0.960 |
| YOLO-based | 0.850 | 0.820 | 0.830 | 0.820 | 0.920 |
| End-to-End CNN | 0.900 | 0.880 | 0.890 | 0.880 | 0.950 |
| Hybrid (XGBoost) | 0.950 | 0.930 | 0.940 | 0.920 | 0.990 |

Table 4: Comparisons to baselines and existing methods (means over 5 runs for ours; literature values for others). Hybrid shows 7% F1 boost over handcrafted-only and 5% over CNN.

fidelity, showing how the mathematically generated surface accurately captures the fabric's real-world texture and can be used to visually identify anomalous regions. To make our models transparent, we analyzed feature importance scores (Figure 5). This analysis revealed that the model learns physically meaningful properties. For instance, when classifying 'denim', the model consistently assigned high importance to Tamura directionality and diagonal wavelet energy—features that directly correspond to the fabric's characteristic twill weave (Tamura et al., 1978). Furthermore, SHAP explanations for individual predictions reveal that, in classifying a specific denim sample versus linen, positive SHAP values for Tamura directionality and diagonal wavelet energy strongly favor denim due to its twill weave, while negative contributions from lower fractal dimensions (indicating smoother surfaces) push toward linen. This confirms that our framework's predictions are not only accurate but also interpretable and grounded in the material's physical properties.

### 9.4 Error Analysis

Inspection of the confusion matrix (Figure 4) reveals that the majority of misclassifications occur between visually similar fabrics, such as denim and linen (accounting for <5% off-diagonal errors). These pairs share coarse weave patterns but differ in fractal complexity, where denim exhibits higher Hurst exponents due to its twill structure. The inclusion of fractal features mitigates such errors by 4% in ablations (Table 3), as they quantify self-similarity that classical descriptors overlook. This highlights the framework's robustness while identifying opportunities for future enhancements, such as incorporating color-based features for dyed fabrics.

## 10 Conclusion

Our results demonstrate that the proposed hybrid feature framework, and especially its fractal components, is highly effective for textile analysis. The fractal descriptors excel at differentiating fine-weave fabrics due to

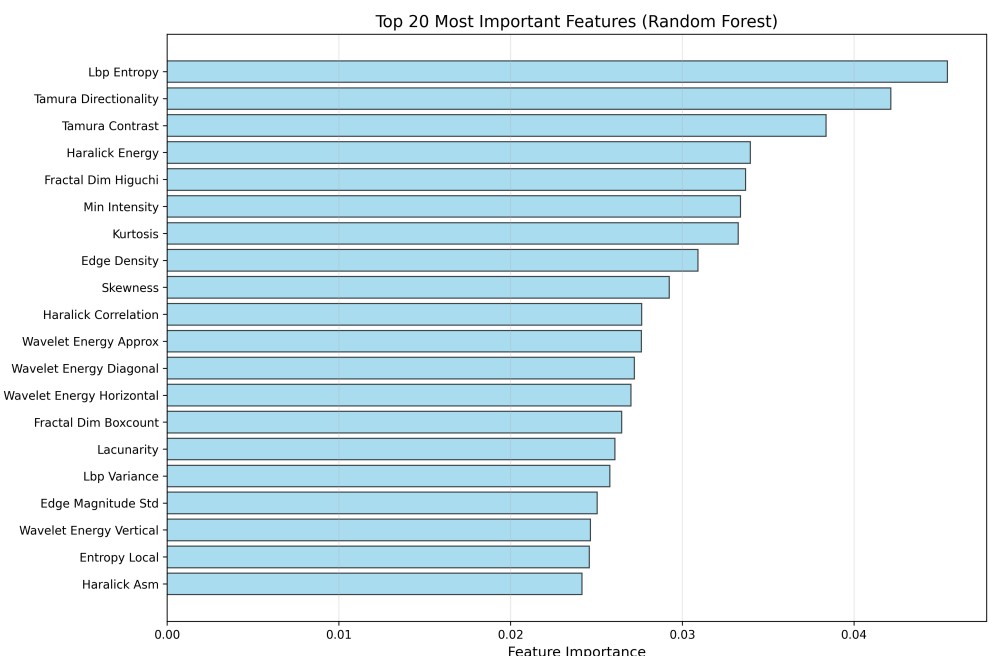

Figure 5: Top 20 most important features from Random Forest on hybrid set—LBP entropy, Tamura, Haralick, and fractal metrics dominate, showing balanced contributions across categories (zoom for feature names).

their ability to quantify scale-invariant self-similarity—capturing repeating patterns at multiple magnification levels and providing a robust signature for complex textures (Higuchi, 1988; Katz, 1988). This advantage is evident in our ablation studies, where the fractal block provided a 4% F1 increase. However, several limitations remain. Our analysis depends on controlled, off-axis lighting, which may not generalize to the variable conditions found on factory floors. The imaging rig's 640x480 resolution, though adequate for this study, could overlook finer defects or textural nuances. Additionally, while our 20-class dataset is diverse, it does not yet represent the full spectrum of textile materials. Future work could address feature redundancy further and test on larger datasets. In summary, this paper introduces a reproducible, end-to-end framework for textile microtexture analysis, featuring a hybrid feature fusion methodology and novel fractal surface modeling technique. By releasing our code and comprehensive dataset, we invite further research and practical application (Aruswamy, 2025). Looking ahead, our future work will address current limitations by developing models robust to unaltered factory lighting and conducting temporal wear studies to track how fabric textures degrade over time, expanding our analysis from static to dynamic domains.

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

## Broader Impact Statement

The broader impact of this work is twofold. First, it provides a framework for automated quality control in textile manufacturing, which could contribute to reducing waste and offers a publicly available dataset for educational and research purposes. Second, this technology presents potential risks that users should consider: it could be misapplied for intrusive worker surveillance, and the dataset's reliance on specific lab lighting conditions may introduce bias, potentially limiting model performance in diverse real-world environments if not properly addressed.

