# OpenReview forum: "Mathematical Modeling and Fractal Geometry for Microtexture Fabric Analysis"
_TMLR — Rejected by TMLR_

### Review · Reviewer_s7R9 · 2025-10-18

**Summary Of Contributions:**

This paper presents a complete, end-to-end pipeline for textile analysis using low-cost, accessible hardware. The main contributions are a hybrid feature engineering approach that combines handcrafted and deep learning features, and a novel fractal overlay module for visualizing surface complexity.

A key strength is the authors' commitment to reproducibility, demonstrated by releasing their code, dataset, and using an accessible Raspberry Pi setup. The fractal analysis module is also an interesting contribution. The paper's primary weakness is the experimental results section, which currently seems incomplete and does not support the central claims.

**Audience:**

Yes

**Audience Explanation:**

The work is a good fit for the TMLR audience. Researchers in applied machine learning, materials science, and interpretable AI would be interested in the accessible pipeline and the novel fractal analysis technique. The focus on creating a reproducible, low-cost solution for a practical problem is valuable to the community.

**Broader Impact Concerns:**

No concerns.

**Claims And Evidence:**

No

**Claims Explanation:**

The conceptual framework of this paper is strong, but the evidence presented in the results section does not currently support the main claims. My concerns are as follows:

The methodology describes several models, including XGBoos, LightGBM, and an attention-based neural network, and notes that XGBoost was the top performer. However, the main results in Table 2 omit these models entirely and report on a different set (SVM, RF, etc.). The results for the best-claimed model should be presented.

In Tables 2 and 3, multiple different models and feature configurations report the exact same performance metrics, identical to three decimal places. This is unlikely and suggests the results may be placeholders.

There are conflicting numbers throughout the text for key experimental details like dataset size (mentioned as both 500 and 2,500) and the number of classes (both 15 and 20).

**Requested Changes:**

The experiments should be re-run to ensure results are final. The tables must be populated with the validated results for all models described in the methodology, especially the top-performing XGBoost model. Additionally, to improve the robustness of results, report the mean and std of performance across multiple different random seeds.

Ensure that all experimental details (e.g., dataset size, class count) are consistent throughout the manuscript.

Captions for figures and tables should be more descriptive, allowing them to be understood as standalone elements. For instance, the caption for Table 2 should state which feature set was used.

---

> ### Author Response · Authors · 2025-12-09
>
> 1. Model Reporting and Table Consistency: All models used in the study (Logistic Regression, MLP, XGBoost, LightGBM, Random Forest, SVM, and the Isolation Forest) are now fully reported. XGBoost remains the top performer, achieving an accuracy of 0.950±0.012 and a Macro F1 of 0.930±0.010. Table 2 has been updated accordingly and now reflects the complete, validated model suite.
>
> 2. Placeholder Values and Statistical Rigor: We acknowledge that earlier versions of Tables 2 and 3 contained placeholder values. All experiments have now been re-run using the finalized pipeline, each trained and evaluated over 5 random seeds.The tables now report means and standard deviations. For example, Random Forest accuracy is 0.940±0.015. In the ablation study, Table 3 shows realistic progression (e.g., F1 increases from 0.700 using only statistical features to 0.920 using all features).
>
> 3. Dataset Size and Class Clarification: Conflicting dataset numbers have been corrected. The dataset contains 20 fabric classes.
> Each class includes 5 swatches with 5 rotated views each, for a total of 500 high-resolution images. All references to “15 classes” or “2,500 images” have been removed and Section 7.1 and the Abstract now state the correct dataset composition.
>
> 4. Descriptive Captions: We have updated all major figure and table captions to be self-contained. Captions now explicitly state the feature set and model context. For example: Table 2 caption clarifies that results are computed on hybrid features, reported as mean ±std over 5 runs. Figure 4 caption now specifies it displays performance across 20 fabrics for Random Forest using hybrid features.

---

### Review · Reviewer_fjmY · 2025-10-27

**Summary Of Contributions:**

This paper presents an end-to-end pipeline for automated textile microtexture analysis using a Raspberry Pi-based imaging system. The main contributions are a hybrid feature engineering approach that combines 41 handcrafted features with deep CNN features from ResNet50 and EfficientNet, a novel interpretable fractal fitting module using Fractional Brownian Motion to generate visual overlays and extract Hurst exponents, and an open dataset of 500 high-resolution fabric images with corresponding feature vectors and fractal visualizations.
The work has notable strengths. The comprehensive feature engineering approach spans multiple texture analysis domains, including statistical, morphological, wavelet, and fractal features. The proposed fractal overlay module offers unique visual and quantitative insights into surface roughness and texture complexity. There is a strong emphasis on reproducibility through public datasets and code release, and the work includes interpretability analysis through SHAP and fractal visualizations. The low-cost, reproducible setup makes the approach accessible for educational and resource-constrained settings.
However, the paper has significant weaknesses that undermine its contributions. The experimental validation is limited, and performance improvements are unclear, with nearly identical metrics across different feature combinations. Comparisons with existing methods are absent, and several claims appear overstated relative to the supporting evidence presented in the results tables.

**Audience:**

Yes

**Audience Explanation:**

Automated textile quality control addresses real industrial needs, and the emphasis on a resource-constrained setup using Raspberry Pi hardware with less than 8GB RAM is relevant for practical deployment scenarios where computational resources are limited. The focus on reproducibility, open data, and code release aligns well with machine learning community interest, even though the execution in this submission is flawed. The systematic exploration of combining handcrafted texture features with deep learning representations could interest researchers working on interpretable machine learning for specialized domains where purely data-driven approaches may be insufficient. The application of fractal geometry to fabric microtexture analysis at this scale represents a systematic documentation of methods that may interest researchers in texture analysis and geometric feature extraction. However, the paper requires substantial revisions to resolve inconsistencies, and support its claims with actual evidence.

**Broader Impact Concerns:**

A proper broader impact statement is present.

**Claims And Evidence:**

No

**Claims Explanation:**

The paper makes several claims that are not adequately supported by the presented evidence. The abstract states there are "consistent improvements in classification F1 scores and defect detection AUC compared to baseline handcrafted feature pipelines," but Tables 2 and 3 show nearly identical performance with F1 equal to 0.940 across all models, and no baseline comparison is actually provided. The claimed improvements from ablation studies, such as a "7% boost" from deep features and "4% increase" from fractal features, are presented as examples with the notation "[e.g., X%]" rather than as actual experimental results.

Section 7.1, 9.1 explicitly states "15 distinct fabric classes," while the abstract and other sections claim "20 fabric types," with the mathematical formula showing "500 images (20 classes × 5 swatches/class × 5 views/swatch)." Figure 6 displays results for 20 classes. This is not a minor discrepancy but a core confusion about the dataset composition.

The defect detection results are inconsistent. The text claims "excellent AUC of [e.g., 0.92]" but Table 2 shows an actual AUC of 0.486, which is worse than random chance. The precision is 1.000, but the recall is only 0.108, indicating the model rarely detects defects. This represents a failure of the defect detection task. The fractal modeling module is presented as a novel contribution, yet Table 3 reveals that adding fractal features maintains the same F1 score (0.940) as other feature sets, providing no evidence of their claimed contribution to classification performance.

No comparisons are provided with existing textile analysis methods despite citing recent work on YOLOv5, YOLOv8, and other state-of-the-art approaches in the related work section. The paper lacks statistical significance tests, confidence intervals, or multiple experimental runs.

**Requested Changes:**

The paper alternates between stating 15 and 20 fabric classes. Please clarify the correct number and update all references consistently across the abstract, sections, figures, and formulas to enable accurate assessment of the experimental scope.

Several key results appear as placeholders "[e.g., X%]" rather than actual values. Please provide the specific F1-score improvements from deep feature fusion and fractal features mentioned in Section 9.1, and clarify the discrepancy between the claimed "excellent AUC of 0.92" and Table 2's reported 0.486.

Table 3 shows identical F1 scores (0.940) across all feature combinations. This suggests either a reporting issue or dataset saturation. Please revise the ablation study to demonstrate differential contributions of each feature group, or discuss why all combinations perform identically.

The paper must compare against existing methods to substantiate improvement claims. Include proper baseline using comparisons with 2-3 existing methods from the related work section such as the YOLO-based approaches or methods from cited papers, and a simple end-to-end CNN trained directly on the images as a modern baseline.

Include multiple experimental runs with different random seeds and report means and standard deviations, provide confidence intervals or standard errors for all reported metrics, and conduct statistical significance tests for claimed improvements. The current single-run results with suspiciously low variance are insufficient.

---

> ### Author Response · Authors · 2025-12-09
>
> 1. Dataset Class Count Clarification: We corrected the inconsistency between “15” and “20” fabric classes. The manuscript now consistently reports 20 classes, verified across all text, equations, tables, and figures. Abstract, Section 7.1, and related experimental discussions were updated to reflect the correct dataset composition.
>
> 2. Replacement of Placeholder Results: All placeholder values (e.g., “X%”) have been replaced with actual experimental results. Macro F1 improved from 0.860 → 0.930 (+7%). Micro-tear AUC improved from 0.700 → 0.780 (+10%). Deep Feature Fusion contributed +7%, and the Fractal Feature Block contributed +4% (confirmed via ablation). Abstract and Section 9.1 now contain complete, statistically supported numbers.
>
> 3. Correction of Identical F1 Scores in Ablation: The previous identical F1 scores (all 0.940) were due to a reporting error. We re-ran the ablations using proper stratified k-fold splits. The new Table 3 shows a logical progression: Macro F1: 0.700→0.920 as feature domains are added. Table 3 and Section 9.1 were rewritten to show accurate incremental contributions.
>
> 4. Defect Detection Metrics Correction: The previously inconsistent defect detection claim (e.g., AUC 0.486 vs. “excellent AUC”) was caused by a stale placeholder and hyperparameter mismatch. We retuned the Isolation Forest model and updated results. Abstract and Table 2 now show the corrected, verified values.
>
> 5. Added Comparisons Against Baselines: We implemented and compared against standard baselines including YOLO-type detectors, End-to-End CNNs, Deep-Only, and Handcrafted-Only models. A new Section 9.2 and Table 4 present the expanded baseline comparisons
>
> 6. Multi-Seed Experiments & Statistical Significance: In response to concerns about single-run variance, we adopted a full 5-seed evaluation protocol. Section 8.3 now details the evaluation protocol; all tables (2, 3, 4) include standard deviations and significance indicators.

---

### Review · Reviewer_oePQ · 2025-11-02

**Summary Of Contributions:**

The paper presents a reproducible framework for automated fabric microtexture analysis using a low-cost Raspberry Pi microscope. It extracts a 41-dimensional handcrafted feature vector that includes statistical, spatial, frequency, and fractal descriptors. A fractal modeling module fits Fractional Brownian Motion surfaces to estimate the Hurst exponent and produce texture roughness maps. The authors also release a dataset of 500 images from various fabric types. Handcrafted features are fused with deep features from pretrained CNNs. Classification results are reported using models such as SVM and MLPs. The authors claim that the hybrid and fractal features improve performance and provide interpretable analysis for industrial and educational applications.

**Audience:**

No

**Audience Explanation:**

The paper offers limited insight, and its findings are unlikely to interest the TMLR audience. For example, Sections 3 and 4 describe basic data acquisition and preprocessing steps that do not provide new methodological contributions. Section 5 appears to simply compile a set of well-known texture features without careful consideration of their relevance or redundancy. From Table 1 and Section 5.1, it seems that mean, standard deviation, and variance are included among the seven statistical features. However, standard deviation and variance give identical information (correlation of 1), and we can also see these in the correlation plot. The same issue applies to Shannon and local entropy, which are both included without an explanation on how they are computed. I believe the feature set was not constructed with sufficient justification and motivation.

For these reasons, I do not believe the paper will be of interest to the broader TMLR audience.

**Claims And Evidence:**

No

**Claims Explanation:**

I am not convinced that the claims in the paper are well supported or that clear evidence is presented. The authors report performance improvements from hybrid handcrafted, deep, and fractal features, but the results do not show a significant gain. Table 3 indicates that simple statistical features perform as well as the full 41-feature set. In addition, there are no quantitative results provided for the deep-features-only baseline.

Several details about the feature design and their relevance are also unspecified. I have additional concerns about the feature definitions and their significance, which I will talk about in my comments below.

Overall, the results and analysis do not substantiate the claimed improvements or interpretability benefits.

**Requested Changes:**

* Clarify and justify the feature design
  - Explain the rationale for including redundant statistical features (e.g., both variance and standard deviation) and multiple entropy measures.
  - Provide details on how Shannon and local entropy are computed, including histogram modeling, neighborhood size, and normalization.
  - Discuss the relevance and independence of each feature category, supported by statistical significance or feature-importance analysis.

- Add additional baselines
  - Include quantitative results for the *deep-features-only* and *handcrafted-only* configurations to support the claim that hybrid fusion improves performance.

- Clarify implementation details
  - Specify how probability distributions for entropy are estimated and how fractal fitting quality is computed.

- Improve figure quality and readability
  - Several figures (e.g., correlation plots) are difficult to read due to small font sizes.

---

> ### Author Response · Authors · 2025-12-09
>
> 1. Redundant Features (Variance vs. Standard Deviation): We agree that variance and standard deviation were redundant. We removed variance from the statistical feature set, reducing the feature vector from 41 to 40 dimensions. Pairwise correlation analysis (Fig. 2) confirmed perfect redundancy, and the remaining features exhibit low cross-domain correlation (<0.5). All counts in the main text (Abstract, Sec. 5) have been updated accordingly.
>
> 2. Entropy Computation Details: Section 5.2 has been expanded to clearly describe the entropy formulations. Shannon entropy is computed over a 256-bin normalized grayscale histogram using H = -Σ (p_i * log2(p_i)) with p_i estimated as normalized counts and smoothed with e = 10^-6. Local entropy uses 9x9 neighborhoods with stride 4 and reflection padding. The section now explicitly outlines histogram modeling, windowing, and normalization procedures.
>
> 3. Feature Relevance and Independence: To justify the feature set, we added a feature-importance analysis using Random Forests and SHAP (Fig. 5). The top discriminative features span multiple domains—e.g., LBP entropy, Tamura directionality, fractal metrics—demonstrating that hybrid features contribute complementary information. Section 9.3 now interprets key features in context.
>
> 4. Missing Baselines (Deep-Only vs. Handcrafted-Only): Section 9.2 now includes a dedicated baseline comparison with a new results table (Table 4). The Hybrid model achieves a Macro-F1 of 0.930, outperforming Handcrafted-Only (0.860) and Deep-Only (0.880), with statistically significant improvements (p<0.05). Table 4 also reports results for YOLO-based and End-to-End CNN baselines.
>
> 5. Fractal Fitting Details: Section 6.2 now clarifies that fractal fitting quality is computed as the R^2 of the linear regression on the log–log curve of length versus interval size. The average fit across all windows is R^2 = 0.93 +- 0.02.
>
> 6. Figure Quality: Figures 2 (correlation matrix) and 3 (PCA/t-SNE) have been regenerated at higher resolution with improved labeling for readability.

---

### Decision · Action_Editor_BTE2 · 2025-12-14

**Recommendation:** Reject

**Additional Comments:**

N/A

**Audience:**

No

**Audience Explanation:**

The overall area (imaging-for-materials-analysis) is probably within scope for a portion of the TMLR audience. But in my judgement the paper reads more like a technical report detailing what the authors did, and less of an in-depth, lasting scientific contribution to the field.

**Claims And Evidence:**

No

**Claims Explanation:**

The authors study an approach for microtexture analysis of textles/fabrics. Specifically, they study a 20-class fabric classification problem given image data (captured using a Raspberry Pi microscope). For this problem they implement a reasonably elaborate pre-processing procedure, followed by aggregating a bunch of statistical, edge, fractal, and other features. They train a bunch of standard ML models on this feature set and report the obtained results.

The paper was met with negative reviews. Several inconsistencies in the text were pointed out. The initial version of the paper contained several missing results (including those corresponding to the best-claimed model, XGBoost). The supposed gains by hybrid fractal and deep features were not deemed to be significant (despite the authors' claims). Moreover, Sections 3, 4, and 5 described lengthy details about the image acquisition and feature processing but were more less standard from the previous literature.

The authors provided responses to some of these concerns (primarily, by including the complete set of results). However, I still feel that the topline claims still aren't supported by evidence. The setup is a good initial step, but to call it a "...a transparent, extensible framework for computational material science, AI driven quality control ..." is a massive over-sell.